# Maternal Pregnancy and Pre-Pregnancy Weight and Behavioural Outcomes in Children

**DOI:** 10.3390/bs14010049

**Published:** 2024-01-12

**Authors:** Berihun A. Dachew, Akilew A. Adane, Rosa Alati

**Affiliations:** 1School of Population Health, Curtin University, Bentley, WA 6102, Australia; rosa.alati@curtin.edu.au; 2enAble Institute, Curtin University, Bentley, WA 6102, Australia; 3Ngangk Yira Institute for Change, Murdoch University, Murdoch, WA 6150, Australia; akilew.adane@murdoch.edu.au; 4Institute for Social Science Research, The University of Queensland, Indooroopilly, QLD 4068, Australia

**Keywords:** ALSPAC, gestational weight gain, maternal BMI, behavioural problems, SDQ

## Abstract

Reported associations of pre-pregnancy weight and/or gestational weight gain with offspring behavioural outcomes are inconsistent. Using data from the Avon Longitudinal Study of Parents and Children (ALSPAC), this study aimed to examine these associations at five developmental stages between the ages of 3 and 16. Over 6800 mother–offspring pairs at age 3 and 3925 pairs at age 16 were included. Pre-pregnancy underweight was associated with a 22% increased risk of total behavioural difficulties (OR = 1.22, 95% CI 1.02–1.45). In separate analyses using the SDQ subscales, pre-pregnancy underweight was linked to a 37% (OR = 1.37, 95% CI 1.14–1.65) and 33% (OR = 1.33, 95% CI 1.01–1.76) increased risk of emotional symptoms and prosocial behaviour problems over time, respectively. While pre-pregnancy overweight was associated with an 11% (OR = 1.11, 95% CI 1.03–1.20) and 18% (OR = 1.18, 95% CI 1.03–1.36) increased risk of conduct and peer relationship problems, respectively, pregnancy obesity was associated with a 43% increased risk of emotional problems (OR = 1.43, 95% CI 1.16–1.77). We found no evidence of associations between gestational weight gain and child behaviour except for a reduced risk in prosocial behaviour problems (OR = 0.82, 95% CI 0.70–0.96). Our findings provide insights into the link between preconception BMI and child behaviour, underscoring the necessity for further research to validate these associations and elucidate underlying mechanisms.

## 1. Introduction

Pre-pregnancy body mass index (BMI) is associated with a range of adverse health outcomes such as miscarriage and placental abruption in mothers, and low birth weight, preterm birth, and other postnatal complications in offspring [1,2,3]. In addition to the physical health consequences, recent meta-analysis findings also revealed positive associations between pre-pregnancy BMI and adverse neurodevelopmental outcomes in offspring, including autism spectrum disorder [4] and attention-deficit/hyperactivity disorder [5].

While gestational weight gain is necessary for healthy fetal development, inadequate or excessive gestational weight gain are also associated with adverse maternal and infant outcomes [6,7]. As a result, the Institute of Medicine (IOM) has set weight gain recommendations based on pre-pregnancy BMI [8,9]. However, existing studies show that up to 70% of pregnant women do not meet the IOM recommendations, with the majority (41% to 64%) gaining above the recommended levels [6,9,10,11].

The intrauterine mechanism through which pre-pregnancy BMI and gestational weight gain might influence child behaviour is only speculative. While it is crucial to acknowledge the intricate interplay of genetic and environmental factors that could contribute to a child’s brain development, existing studies indicate that suboptimal pre-pregnancy BMI and gestational weight gain affect fetal brain development through intestinal flora, oxidative stress, inflammation-induced mal-programming, and leptin resistance mechanisms [12,13,14]. Therefore, it is possible that extremes of pre-pregnancy BMI and gestational weight gain may impact fetal brain development, thereby influencing behavioural outcomes later in life.

Despite several studies examining the association between pre-pregnancy BMI and adverse physical and neurodevelopmental outcomes [15,16,17,18], studies on the association between pre-pregnancy BMI and offspring behavioural problems are limited, and the findings are inconsistent. While some studies have reported an increased risk for emotional and behavioural outcomes in the offspring of mothers who are underweight, overweight, or obese [19,20,21], others have found little or no evidence of associations [22,23,24]. Similarly, we are only aware of two studies that examined the association between gestational weight gain and child emotional and behavioural problems, with one reporting a positive association between excessive gestational weight gain and child behaviour [25], while the other found no associations [26].

Most existing studies reporting the impact of pre-pregnancy BMI and gestational weight gain on offspring behavioural problems were based on small sample sizes [19,20,25,26], examined the associations at one time-point only [20,21,22,23,24,25,26], and did not consistently account for important confounding factors, such as maternal education, smoking, and psychopathology [19,21,22,25]. In this study, we aimed to address these gaps in the literature by examining the associations at five developmental periods ranging from age 3 to 16 years. We availed ourselves of the Avon Longitudinal Study of Parents and Children (ALSPAC), a large population-based prospective birth cohort study with comprehensive data to account for a range of known confounders. We also used paternal BMI as a negative control exposure to determine the plausibility of intrauterine effects of maternal pre-pregnancy BMI. This can be done by comparing the strength of association between maternal and paternal BMI with offspring behavioural outcomes. If pre-pregnancy BMI is involved in the causal pathway to offspring emotional and behavioural problems, then the association will be more strongly seen with maternal, not paternal, BMI.

## 2. Methods

### 2.1. Study Participants

Our study sample comprised participants in the ALSPAC cohort, an ongoing population-based longitudinal birth cohort in Bristol, Avon, United Kingdom. The core enrolled sample consisted of 14,541 pregnancies of women residents in the former county of Avon, United Kingdom, who had expected delivery dates between 1 April 1991 and 31 December 1992 [27,28,29]. Of the 14,062 live births, 13,617 were singletons and alive at one year. Since recruitment, the parents and children have been followed up with regularly via questionnaires and clinic assessments. In this current study, term (birth that occurs between the 37th and 42nd weeks of gestation) and normal birth weight children (weighing between 2500 and 4000 g) who had complete data on the exposure (pre-pregnancy BMI and gestational weight gain) and outcome (child emotional and behavioural problem measures) variables and covariates were included in the final analyses (Figure 1). Overall, the ALSPAC cohort consists of participants from a predominantly white population in the Avon region of the UK. Although, on average, participants had higher socioeconomic position indicators than the general population of women in both Avon and Britain, the ALSPAC mothers were slightly more likely to be living in overcrowded conditions (on average more than one person per room) than either the average Avon or British woman. Further details on the ALSPAC cohort profile, including information on recruitment, study design, and generalisability, have been previously reported [27,29]. Additionally, the study website contains information on all the available data through a fully searchable data dictionary and variable search tool (http://www.bristol.ac.uk/alspac/researchers/our-data (accessed on 8 July 2023)).

All methods were performed in accordance with the relevant guidelines and regulations. Ethical approval for the ALSPAC study was obtained from the ALSPAC Ethics and Law Committee as well as the Local Research Ethics Committees. Written informed consent for the use of data collected via questionnaires and clinics was obtained from participants following the ALSPAC Ethics and Law Committee’s recommendations at the time. The details on the ethical approval, including the dates of approval and associated reference numbers, can be found on the ALSPAC website [30].

### 2.2. Exposure Measures

#### 2.2.1. Pre-Pregnancy BMI

Mothers reported their pre-pregnancy weight and height on self-completed questionnaires administered around the 18th week of gestation. This information was used to calculate pre-pregnancy BMI (calculated as weight in kilograms divided by height in metres squared). Reported maternal weight was strongly correlated with weight measured at the first antenatal clinic visit (median gestational age: 10 weeks [interquartile range, IQR: 8–14]) (r = 0.94, *p* < 0.001). Paternal BMI was derived from self-reported height and weight from a questionnaire sent to partners at 18 weeks gestation. Maternal and paternal BMI were grouped according to the World Health Organization categories for underweight (<18.50 kg/m^2^), normal (18.50–24.99 kg/m^2^), overweight (≥25.00 kg/m^2^), and obese (≥30.00 kg/m^2^) [31].

#### 2.2.2. Gestational Weight Gain

Weight was measured routinely at all antenatal clinic visits at the time of data collection. Six trained midwives abstracted data from obstetrical medical records. Data included every measurement of weight entered and the corresponding gestational age and date for the time of measurement. There were no between-midwife variations in mean values of abstracted data and repeat data-entry checks demonstrated error rates consistently below 1%. The first weight measurement (kg) was subtracted from the last to determine absolute weight gain, derived for all women who had at least one weight measurement before 18 weeks of gestation and one after 28 weeks of gestation. The first weight measurement was collected at a median of 10 weeks (IQR: 8, 11), and the last weight measurement was collected at a median of 39 weeks (IQR: 38, 40). The time between the last weight measurement and delivery was brief (median: 0 weeks; IQR: 0, 1). Women were categorised as having inadequate, adequate, or excessive gestational weight gain based on the 2009 IOM recommendations [9]. According to the IOM (2009) guideline, the recommended range of weight gain is 12.5–18 kg for underweight, 11.5–16 kg for normal-weight, 7–11.5 kg for overweight, and 5–9 kg for obese singleton pregnant women. For twin pregnancies, the IOM recommends a gestational weight gain of 17–25 kg for normal-weight women, 14–23 kg for overweight women, and 11–19 kg for obese women. Women gaining less than these ranges within each BMI category are classified as having inadequate weight gain, and those gaining more as having excessive gestational weight gain.

### 2.3. Outcome Measures

Behavioural problems were measured using the Strengths and Difficulties Questionnaire (SDQ), completed by parents, usually mothers, when the child was approximately 3.5, 7, 9, 11, and 16 years old [32]. The SDQ has previously been found to be a valid and reliable instrument used to screen behavioural problems in children aged 3–16 years, and has been widely used in research and clinical practice [32,33]. The tool comprises twenty-five questions with five sub-scales: hyperactivity/inattention problems, emotional symptoms, conduct problems, peer relationship problems, and prosocial behaviour (each containing five items). Scores for the sub-scales range from 0 to 10, and the first four subscales are combined to calculate a total difficulties score, ranging from 0 to 40. A higher score indicates higher symptoms, except for prosocial behaviour, where a lower score indicates more difficulties [32]. Children were categorised as ‘borderline/abnormal’ if they scored in the ‘abnormal’ or ‘borderline ‘ranges, according to cut-offs suggested by Goodman for total behavioural difficulties and for each SDQ sub-scale [33]. Binomial cut-off: total difficulties (≥14), hyperactivity/inattention problems (≥6), conduct problems (≥3), emotional symptoms (≥4), peer relationship problems (≥3), and prosocial behaviour (≤5).

### 2.4. Covariates

Potential confounders were selected based on previous reports of their association with pre-pregnancy BMI and/or gestational weight gain and offspring emotional and behavioural problems [34,35,36,37,38,39,40]. These include maternal age, education, marital status, parity, maternal smoking and alcohol use during pregnancy, maternal antenatal anxiety and depressive symptoms, and offspring sex. Maternal depressive symptoms was measured at 32 weeks of gestation using the Edinburgh Postnatal Depression Scale (EPDS). Scale scores were dichotomised using the recommended cut-off score for depression (12 out of 30) [41]. Symptoms of antenatal anxiety at 32 weeks of gestation were measured with the Crown-Crisp Experiential Index (CCEI), and a score of ≥8 was used to indicate clinically significant anxiety symptoms [42]. Data on these potential confounders were obtained from obstetric records and questionnaires administered during pregnancy.

### 2.5. Statistical Analysis

First, we used descriptive statistics such as proportions and means with standard deviations (SDs) to describe the sociodemographic and clinical characteristics of participants included in this study. Then, we conducted univariable and multivariable Generalized Estimating Equation (GEE) analyses to investigate the associations between pre-pregnancy BMI and gestational weight gain, and behavioural problems in offspring over time, computing odds ratios (ORs) and 95% confidence intervals (CIs) as measures of association. Additionally, we examined the potential interaction between pre-pregnancy BMI and/or gestational weight gain and sex on the risk of behavioural outcomes in children. We then used paternal BMI as a negative control for intrauterine exposure and compared the association between maternal and paternal BMI with offspring behavioural outcomes. We also examined the associations at each developmental period (i.e., 3.5, 7, 9, 11, and 16 years of age) using a logistic regression analysis. We primarily used the categorical SDQ scores, comparing the ‘borderline or abnormal group’ with the ‘normal group’, and the continuous SDQ scores were used as sensitivity analyses. A sample with complete data across all exposure, outcome, and confounding variables was used to examine the associations. To account for missing data, we conducted sensitivity analyses using multivariate multiple imputations by chained equations. We used 50 cycles of regression switching and generated 50 imputed datasets. All covariates included in the regression model and additional auxiliary variables predictive of incomplete variables were included in the regression model and imputed, and the analyses were repeated. All statistical analyses were conducted using Stata 16 software [43].

## 3. Results

Table 1 shows the sociodemographic and clinical characteristics of the mothers and children included in this study. Among the 10,571 mothers with complete data on pre-pregnancy BMI, 5.6% were obese, 15.2% were overweight, and 4.9% were underweight, with a mean (SD) pre-pregnancy BMI of 23.0 (3.85) kg/m^2^. Of the 10,066 mothers with complete data on gestational weight gain, 33.3% gained weight below the IOM recommendations, while 27.6% gained weight above the IOM recommendations, with a mean (SD) gestational weight of 12.7 (4.64) kg. Over three-quarters (77.3%) of mothers were married, 23.0% smoked tobacco, 15.7% consumed alcohol during pregnancy, and 18.4% experienced antenatal depressive symptoms. Approximately half (51%) of the children were male, with a mean (SD) birth weight of 3.5 (0.45) kg.

We also compared the characteristics of mothers and children with data on total behavioural difficulties to those with missing data on this outcome. In comparison with those included in the analyses, mothers of children who were excluded due to loss to follow-up or missing data were never married, had lower levels of education, were multiparous, smoked tobacco, drank alcohol, and experienced more antenatal depressive and anxiety symptoms (Appendix A).

Table 2 shows the proportion of children classed as having behavioural problems, using the Goodman cut-offs, with total behavioural difficulties and each SDQ subscale from age 3 to 16 years stratified by sex. Overall, more boys than girls had total behavioural difficulties, hyperactivity/inattention problems, conduct problems, peer relationship problems, and problems with prosocial behaviours, except at the age of 16, in which total behavioural difficulties and conduct problems were more prevalent among girls than boys (*p* < 0.05). In contrast, the proportion of children scoring in the ‘borderline/abnormal’ range for emotional problems was higher for girls than boys (*p* ≤ 0.01), though the difference was not statistically significant at the age of 3 (*p* = 0.11). No evidence of interaction between pre-pregnancy BMI or gestational weight gain and gender was found in all age groups (*p* ≥ 0.07 for gestational weight gain and ≥0.16 for pre-pregnancy BMI).

Mean scores for each SDQ sub-scale and total difficulties are shown in Appendix A. The mean (SD) score of total difficulties was 12.5 (5.7) at age 3 and 6.2 (4.8) at age 16. While the mean SDQ scores for total difficulties, hyperactivity/inattention problems, and conduct problems decreased as the offspring’s age increased, emotional symptoms, peer relationship problems, and prosocial behaviours remained stable.

Table 3 shows univariable and multivariable associations between pre-pregnancy BMI and gestational weight gain, and offspring emotional and behavioural problems over five developmental periods (3.5, 7, 9, 11, and 16 years). Our unadjusted GEE model showed that pre-pregnancy underweight was associated with a 38%, 27%, 18%, and 53% increased risk of total behavioural difficulties, hyperactivity/inattention problems, conduct problems, and emotional symptoms over time, respectively. Pre-pregnancy overweight was associated with 12%, 13%, and 20% increased risk of total behavioural difficulties, conduct problems, and peer relationship problems over time, respectively. Pre-pregnancy obesity was associated with a 48% increased risk of peer relationship problems and a 38% reduction in the odds of having prosocial behaviour problems over time. In the adjusted GEE model, pre-pregnancy underweight was associated with a 22% (OR = 1.22, 95% CI 1.02–1.45) increased risk of total behavioural difficulties, a 37% (OR = 1.37, 95% CI 1.14–1.65) increased risk of emotional symptoms, and 33% (OR = 1.33, 95% CI 1.01–1.76) increased risk of prosocial behaviour problems over time. The observed association between pre-pregnancy underweight and externalising behaviours (hyperactivity/inattention problems and conduct problems) did not exist after adjustment. Pre-pregnancy overweight was associated with an 11% and 18% of increased risk of conduct and peer relationship problems, respectively. The association between pre-pregnancy overweight and total behavioural difficulties that we observed in the unadjusted model was attenuated after accounting for confounders. We also found 43% increased risk of peer relationship problems (OR = 1.43, 95% CI 1.16–1.77) and a 38% (OR = 0.62, 95% CI 0.44–0.86) reduced risk of prosocial behaviour problems over time in offspring of mothers with pre-pregnancy obesity. We found no evidence of associations between gestational weight gain and child behaviour except for excessive gestational gain, which was associated with an 18% reduction in prosocial behaviour problems in the adjusted model (OR = 0.82, 95% CI 0.70–0.96). Broadly consistent results were obtained when we reran the analyses using the continuous SDQ scores (Appendix A) and on the imputed data sets (Appendix A).

We also examined the effect of pre-pregnancy BMI and gestational weight at each developmental period (Appendix A). We found that pre-pregnancy BMI and gestational weight gain were not associated with total emotional and behavioural difficulties in children across all age groups. However, further analyses for each SDQ subscale revealed few associations. At age 3, pre-pregnancy overweight was associated with conduct problems (OR = 1.21, 95% CI 1.05–1.39). Pre-pregnancy underwight (OR = 1.36, 95% CI 1.06–1.76) and excessive gestational weight gain (OR = 1.16, 95% CI 1.01–1.33) were also associated with emotional problems; however, these associations did not exist after adjusting for confounding factors.

At age 7, pre-pregnancy underweight was associated with emotional problems (OR = 1.65, 95% CI 1.20–2.28). At age 9, pre-pregnancy obesity (OR = 1.78, 95% CI 1.31–2.41), excessive (OR = 1.22, 95% CI 1.01–1.47) and inadequate gestational weight gain (OR = 1.24, 95% CI 1.03–1.49) were associated with an increased risk of peer relationship problems. Pre-pregnancy obesity (OR = 0.38, 95% CI 0.18–0.77) and excessive gestational weight gain (OR = 0.71, 95% CI 0.54–0.92) were associated with fewer difficulties in prosocial behaviour.

While pre-pregnancy underweight was associated with emotional problems at age 11 (OR = 1.49, 95% CI 1.02–2.18), excessive gestational weight gain was associated with a reduction in the odds of having prosocial behaviour problems (OR = 0.63, 95% CI 0.47–0.84). At age 16, pre-pregnancy underweight was associated with emotional problems (OR = 1.74, 95% CI 1.16–2.60) and hyperactivity/inattention problems (OR = 1.96, 95% CI 1.27–3.05). No other significant associations were observed. The observed associations between pre-pregnancy underweight and emotional problems at age 11 and 16 years were not replicated in the imputed data at each time-point (Appendix A). Unlike the complete cases analysis results, using the imputed dataset, pre-pregnancy overweight (OR = 1.24; 95% CI, 1.03–1.48) and obesity (OR = 1.51; 95% CI, 1.10–2.05) were associated with greater peer relationship problems at age 16 years. All other results were broadly consistent with the results of the complete case analyses (Appendix A).

### Maternal vs. Paternal BMI

Our GEE model found that paternal obesity was associated with 1.35 fold increased risk of hyperactivity/inattention problems over time (OR = 1.35; 95% CI, 1.09–1.67). We also found a 36% increased risk of peer relationship problems in offspring of fathers with obesity (OR = 1.36; 95% CI, 1.10–1.68). This association remained, but the magnitude reduced after adjustment to maternal pre-pregnancy BMI (OR = 1.29; 95% CI, 1.03–1.61) (Appendix A). On the other hand, the association between maternal pre-pregnancy overweight and obesity, and peer relationship problems remain unchanged after adjusting for paternal BMI (Appendix A).

In a logistic regression at each time-point, paternal obesity was associated with peer relationship problems at age 9 (OR = 1.43, 95% CI 1.03–1.98) and age 11 (OR = 1.58, 95% CI 1.13–2.21), and hyperactivity/inattention problems at age 9 (OR = 1.52, 95% CI 1.07–2.14). Paternal overweight was associated with prosocial behaviour (OR = 0.74, 95% CI 0.57–0.95) at age 9, and paternal underweight was not found to be associated with any of the SDQ subscales at all ages (Appendix A). Overall, the associations between maternal pre-pregnancy BMI and offspring behavioural problems were of greater magnitude than paternal BMI when associations were observed, though parental BMI did not appear to be statistically significant in most cases. The maternal associations also remained stronger than the paternal associations after mutual the adjustment of maternal and paternal BMI.

## 4. Discussion

In this longitudinal population-based pregnancy cohort study, we found that pre-pregnancy underweight was associated with a 22% increased risk of total behavioural difficulties. We found no evidence of associations between pre-pregnancy overweight or obesity and gestational weight gain, and total behavioural difficulties in children over time and at each developmental period (3.5, 7, 9, 11, and 16 years). In separate analyses using the SDQ subscales, pre-pregnancy underweight was associated with 37% and 33% increased risk of emotional symptoms and prosocial behaviour problems over time, respectively. However, the observed association between pre-pregnancy underweight and hyperactivity/inattention problems, as well as conduct problems, in the unadjusted model, did not persist after accounting for confounders, such as maternal age, education, marital status, parity, maternal smoking and alcohol use during pregnancy, and maternal antenatal anxiety and depressive symptoms, underscoring the role of confounders in the association. While pre-pregnancy overweight was associated with an 11% and 18% increased risk of conduct and peer relationship problems over time, respectively, pregnancy obesity was associated with a 43% increased risk of emotional problems and a 38% reduced risk of prosocial behaviour problems. We found no evidence of associations between gestational weight gain (excessive or inadequate) and child behaviour except for prosocial behaviour problems.

In the EDEN birth cohort study, Dow et al. examined the association between maternal pre-pregnancy BMI and child hyperactivity/inattention symptoms (HIS) at 3, 5, and 8 years of age, and found that maternal pre-pregnancy obesity, but not overweight, was associated with an increased likelihood of a high HIS trajectory in children [44]. We found no other similar study that examined the associations at multiple time-points. However, our findings at each time-point are broadly consistent with the existing studies that examined the associations between pre-pregnancy BMI and child behaviour measured using SDQ or CBCL measures [20,21,23,24,45]. In a prospective cohort study, Robinson et al. [24] reported a positive association between pre-pregnancy obesity and hyperactivity/inattention problems at age 7–8 years. However, no evidence of association was found between pre-pregnancy overweight and total behavioural difficulties with other SDQ subscales.

In another longitudinal study, Jo et al. [20] reported that pre-pregnancy obesity (II/III) was associated with emotional and peer relationship problems at age 6 but not with hyperactivity/inattention problems, conduct problems, and prosocial behaviours. Pre-pregnancy overweight and obesity (I) were not found to be associated with any of the SDQ subscales. In a Danish National Birth Cohort study, Mikkelsen et al. [21] also examined the associations at age 7, and reported associations between maternal pre-pregnancy overweight and obesity, and child behaviour (for total and each SDQ subscales) [21]. Our study found no evidence of an association between pre-pregnancy overweight and obesity, and child behaviour at age 7. At age 9, however, we found a 78% increased risk of peer relationship problems in children of mothers with pre-pregnancy obesity compared to mothers with normal pre-pregnancy BMIs. Consistent with our findings, Dow et al. also found increased odds of a high peer relationship score in children of mothers with maternal pre-pregnancy obesity [46].

In two pregnancy cohorts (Generation R and ALSPAC), Brion et al. [22] also explored the effect of pre-pregnancy BMI on child cognition and behaviour, and reported mixed findings. In that study, pre-pregnancy overweight was associated with child total behaviour problems and conduct problems in the Generation R cohort but not in the ALSPAC cohort. The study reported no evidence of associations of maternal overweight with child attention and emotional problems in both cohorts. However, Brion et al.’s [22] study examined the associations at an early age (3–4 years). In addition, the study neither separately assessed the effect of overweight and obesity, nor did it examine the effect of pre-pregnancy underweight and gestational weight gain.

Although a few studies have examined the association between high maternal pre-pregnancy BMI (overweight or obese) and offspring behavioural problems, the link between pre-pregnancy underweight and child behaviour is rarely studied. Using three French birth cohorts (EDEN, ELFE, and EPIPAGE), Dow et al. examined the link between pre-pregnancy underweight and peer relationship problems in offspring at 5 years of age [46]. While pre-pregnancy underweight was associated with a 30% increased risk of peer relationship problems in the ELFE cohort, no evidence of an association was observed in the EDEN and EPIPAGE-2 cohorts [46]. Dow et al. also reported no evidence of an association between pre-pregnancy underweight and hyperactivity-inattention problems [44,47] using the same cohorts. In a study by Jo et al.’s [20], pre-pregnancy underweight was not associated with child behaviour symptoms, including emotional problems, at the age of 6 years. In contrast, we found constituent (albeit moderate) associations between maternal pre-pregnancy underweights and child emotional problems, with a slightly increased risk with age.

The observed association between pre-pregnancy underweight and adverse child behaviour may be attributable to several mechanisms, such as inadequate prenatal micronutrient status, oxidative stress, low birth weight, and/or preterm birth [48,49,50]. Underweight women were at higher risk of vitamin and mineral deficiencies, including iron, zinc, vitamin A, and iodine, than normal-weight women [48], and there is existing evidence on the association between micronutrient deficiency and adverse child behaviour [51]. Considering the crucial role of nutrition in the early stages of fetal development, adverse nutritional deprivation in this sensitive period of development likely leads to a poor foundation of brain structures, affecting the development of cognitive, motor, and emotional skills in childhood [52]. Maternal underweight is also a known risk factor for low birth weight and preterm birth, [50] which are predictors of adverse brain development [53], supporting the possibility that pre-pregnancy underweight may influence child behaviour. Previous studies also suggest the detrimental influence of pre-pregnancy underweight on the physical and intellectual development of the child [15,54,55,56,57]. It is worth noting that our study was based on term and normal birth weight children, and the inclusion of preterm and low birth weight children in our analysis provided consistent results (Appendix A), suggesting that low birth weight and preterm birth did not explain the observed associations.

In this current study, we also found associations between paternal obesity (measured around the time of pregnancy of the index child) and child hyperactivity/inattention problems and peer relationship problems. Existing studies reported mixed results on the association between paternal BMI and behavioural outcomes in children. While Robinson et al. [24] found no evidence of associations, Mikkelsen et al. [21] reported positive associations between paternal obesity and offspring behavioural problems. Similar to our findings, Mikkelsen and colleagues also reported stronger associations with maternal pre-pregnancy obesity compared with paternal obesity [21].

In this study, we also examined the association between gestational weight gain and child behavioural outcomes. We found that the offspring of mothers who gained weight below and above the IOM recommendations were at increased risk of peer relationship problems at age 9. Excessive gestational weight gain was inversely associated with prosocial behaviour at ages 9 and 11. Pugh et al. [26] and Tore et al. [25] examined the association between gestational weight gain and child behavioural problems at ages 7 and 10, respectively. While Tore et al. [25] reported a positive association between excessive gestational weight gain and child behaviour, Pugh et al. [26] found no associations. It is important to note that these studies were based on a small sample (*n* ≤ 511), assessed the outcomes at one time-point only, and did not account for important confounders. Our study was able to account for these limitations—we used a much larger sample of mothers and children, were able to account for a range of known confounders, including maternal psychopathology, and were able to assess the outcomes at five developmental points. Huang et al. also reported a link between gestational weight gain and adverse neurobehavioural development; however, this study did not examine the association beyond 12 months of the offspring’s age [58].

The inverse association between pre-pregnancy obesity and excessive gestational weight gain, and prosocial behaviour in offspring that we have observed in this study warrants further investigation. While pre-pregnancy obesity reduced the risk of prosocial behaviour problems by 38%, excessive gestational weight gain showed an 18% reduced risk of prosocial behaviour problems over time. We are only aware of two previous studies that examined the association between pre-pregnancy BMI and/or gestational weight gain, and prosocial behaviours in offspring [20,24]. Although the associations did not reach agreed standards for statistical significance, Robinson et al. [24] reported a similar degree of association in children aged 7–8. Jo et al. [20] also examined the association between pre-pregnancy obesity and prosocial behaviours in offspring at age 6, and found no evidence of associations. Additional evidence is needed to draw definite conclusions on the association.

The prospective design, large sample size, and long follow-up period of this study allowed us to examine the effect of each pre-pregnancy BMI category (underweight, overweight, and obese) separately and measure offspring behavioural problems at several developmental time-points, which were the limitations of most existing studies. The population-based samples minimised the possibility of selection bias. We also accounted for a range of known confounders and used paternal BMI as a negative control exposure to determine the plausibility of the intrauterine effects of maternal pre-pregnancy BMI.

The following limitations should also be considered. Children’s behavioural problems were reported by parents, which might introduce bias into our findings. Additionally, there may be recall and measurement biases, as this study relied on self-reported height and weight measures. However, self-reported maternal weight showed a strong correlation with the weight measured at the first antenatal clinic visit (r = 0.94), and existing evidence has demonstrated that self-reported height and weight are highly correlated with objectively measured values [59]. We also relied on symptoms rather than diagnostic assessments. This may lead to random measurement errors. However, SDQ is a valid and reliable tool for assessing emotional and behavioural problems in children and adolescents, and has been widely used in research and clinical practice [32]. While our data allowed us to adjust for a wide range of confounding factors, the possibility of residual or unmeasured confounding factors is still likely, encompassing variables such as parenting style, socioeconomic status, maternal diet and physical activity, environmental stressors, and factors related to genetics, family dynamics, and maternal mental health beyond the prenatal period. Attrition may also compromise the generalisability of the findings and may potentially bias estimates. However, previous work in ALSPAC has suggested that selective drop-out does not bias the prediction of risk of behavioural disorders [60]. Moreover, estimates from multiple imputations and complete case analyses were broadly comparable, suggesting attrition due to missing data was unlikely to have biased our results.

Our findings have public health, clinical practice, and policy implications. The observed association between pre-pregnancy underweight and the risk of emotional problems in children highlights the importance of achieving the right weight prior to pregnancy. Healthcare providers can provide personalised counselling to women of reproductive age, addressing the potential impact of pre-pregnancy weight on children’s emotional well-being. This may involve discussing lifestyle factors, nutrition, and exercise to support a healthy preconception period. Routine screening for emotional and behavioural problems for the children of mothers who were underweight before pregnancy might also help to mitigate more severe sequelae. While we found weak and inconsistent associations between pre-pregnancy overweight or obesity and gestational weight gain, and child behavioural outcomes, encouraging healthy weight management practices and lifestyle choices before and during pregnancy should remain part of comprehensive maternal care for the myriad of outcomes.

## 5. Conclusions

Pre-pregnancy underweight was found to be associated with emotional problems in children. Public health initiatives should emphasise the importance of achieving a healthy weight before pregnancy in order to mitigate the risk of emotional problems in children. The weak and inconsistent evidence of associations between pre-pregnancy overweight, obesity, and gestational weight gain, and child behavioural outcomes underscores the need for further studies using contemporary data to validate these associations and elucidate underlying mechanisms.

## Figures and Tables

**Figure 1 behavsci-14-00049-f001:**
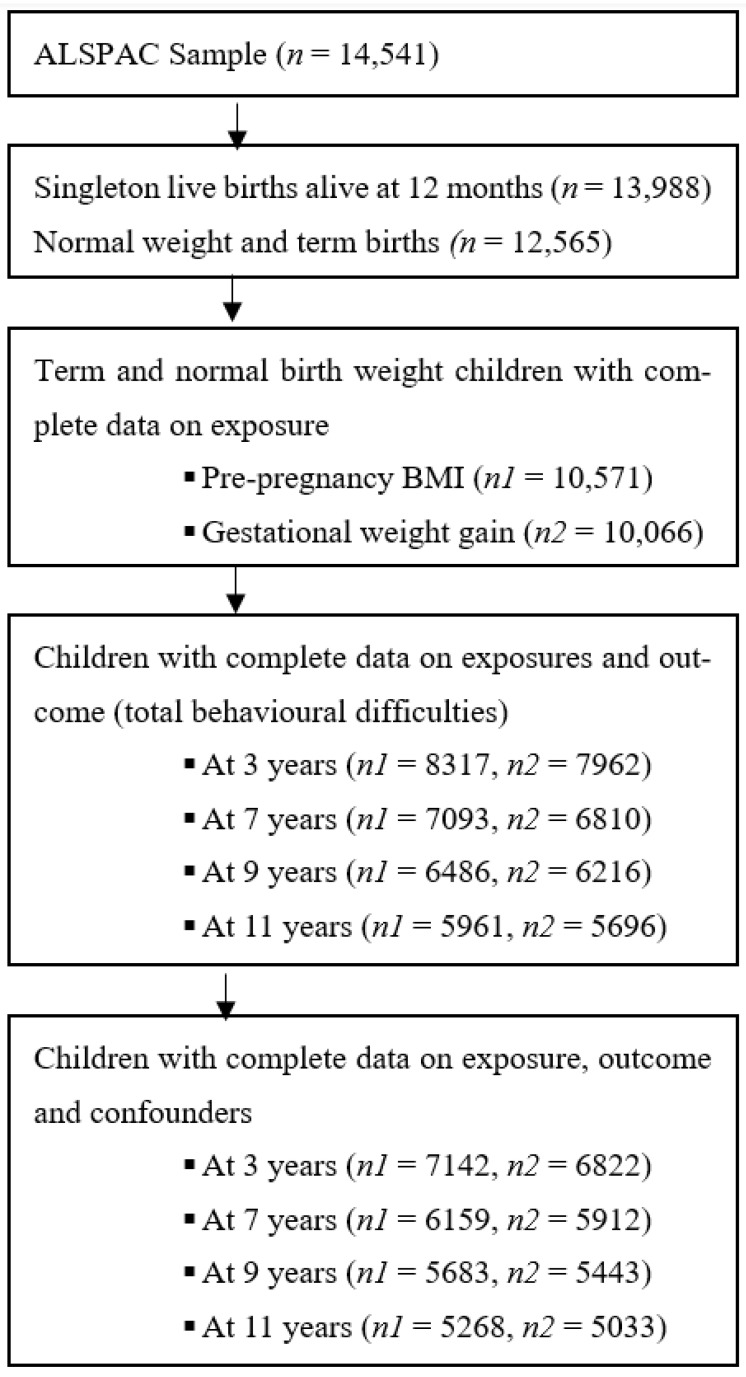
Study participants included in the analysis.

**Table 1 behavsci-14-00049-t001:** Characteristics of study participants.

Characteristics	*n* (%) $ or Mean (SD)				
	Total (*n* = 10,571)	Under Weight (*n* = 520)	Normal Weight (*n* = 7855)	Overweight (*n* = 1607)	Obese (*n* = 589)
Maternal age at delivery (mean, SD)	28.0 (4.8)	26.2 (5.3)	28.1 (4.8)	28.0 (4.7)	27.7 (4.7)
Maternal education #					
Certificate of secondary education	1320 (13.8)	92 (20.6)	859 (12.0)	254 (17.7)	115 (22.5)
Vocational	961 (10.1)	50 (11.2)	682 (9.5)	154 (10.8)	75 (14.7)
O level	3530 (37.0)	154 (34.5)	2620 (36.7)	562 (39.3)	194 (38.0)
A level	2347 (24.6)	100 (22.4)	1818 (25.4)	338 (23.6)	91 (17.8)
Degree	1381 (14.5)	51 (11.4)	1170 (16.4)	124 (8.7)	36 (7.0)
Marital status					
Married	7977 (77.3)	324 (64.9)	5934 (77.3)	265 (80.6)	454 (78.7)
Never married	1759 (17.0)	146 (29.3)	1292 (16.8)	226 (14.4)	95 (16.5)
Widowed/divorced/Separated	585 (5.7)	29 (5.8)	449 (5.9)	79 (5.0)	28 (4.8)
Gestational weight gain, IOM category					
Inadequate (less than recommended)	3176 (33.4)	249 (54.6)	2543 (36.1)	255 (17.4)	129 (23.7)
Adequate (recommended)	3724 (39.1)	162 (25.5)	2951 (41.8)	482 (33.0)	129 (23.7)
Excessive (more than recommended)	2615 (27.5)	45 (9.9)	1559 (22.1)	725 (49.6)	286 (52.6)
Parity					
Nullipara	4489 (44.2)	227 (46.4)	3459 (45.7)	614 (39.6)	189 (33.6)
Multipara	5674 (55.8)	262 (53.6)	4102 (54.3)	936 (60.4)	374 (66.4)
Alcohol drinking in pregnancy					
Yes	1610 (15.7)	67 (13.5)	1236 (16.2)	242 (15.4)	65 (11.3)
No	8667 (84.3)	429 (86.5)	6396 (83.8)	1333 (84.6)	509 (88.7)
Smoking during pregnancy					
Yes	2374 (23.0)	168 (33.50)	1695 (22.1)	376 (23.8)	135 (23.4)
No	7963 (77.0)	334 (66.5)	5986 (77.9)	1202 (76.2)	441 (76.6)
Maternal antenatal anxiety symptoms					
Yes	2122 (22.1)	134 (29.3)	1548 (21.6)	315 (21.7)	125 (24.3)
No	7469 (77.9)	323 (70.7)	5623 (78.4)	1133 (78.3)	390 (75.7)
Maternal antenatal depressive symptoms					
Yes	1811 (18.4)	122 (26.0)	1277 (17.5)	278 (18.5)	134 (24.9)
No	8016 (81.6)	347 (74.0)	6042 (82.6)	1222 (81.5)	405 (75.1)
Child sex					
Male	5388 (50.9)	252 (48.5)	4013 (51.1)	817 (50.8)	306 (52.0)
Female	5183 (49.1)	268 (51.5)	3842 (48.9)	790 (49.2)	283 (48.0)
Gestational age at delivery in weeks (mean, SD)	39.8 (1.3)	39.7 (1.30)	39.8 (1.3)	39.8 (1.3)	39.9 (1.3)
Birth weight in kg (mean, SD)	3.5 (0.45)	3.3 (0.39)	3.5 (0.44)	3.6 (0.47)	3.7 (0.51)

Abbreviations: BMI, body mass index (calculated as weight in kilograms divided by height in metres squared); IOM, Institute of Medicine; SD, standard deviation. # O level indicates examinations taken and passed at 16 years of age; A level, examinations taken and passed at 18 years of age on leaving secondary school. $ percentage refers to column percentage.

**Table 2 behavsci-14-00049-t002:** Proportion of borderline/abnormal SDQ scores by age and sex.

Emotional and Behavioural Problems	Sex	3 Years (*n* = 9207)	7 Years (*n* = 7748)	9 Years (*n* = 7447)	11 Years (*n* = 6811)	16 Years (*n* = 5202)
Total behavioural difficulties (*n*, %)	Male	1970 (41.5)	489 (12.3)	445 (11.9)	404 (11.9)	189 (7.6)
	Female	1611 (36.1)	323 (8.6)	298 (8.0)	250 (7.3)	247 (9.1)
	Total	3581 (38.9)	812 (10.5)	743 (9.9)	654 (9.6)	436 (8.4)
Hyperactivity/inattention problems (*n*, %)	Male	393 (8.3)	929 (23.4)	674 (17.2)	555 (16.4)	310 (12.3)
	Female	255 (5.7)	477 (12.6)	342 (9.2)	253 (7.4)	199 (7.3)
	Total	648 (7.0)	1517 (18.5)	989 (13.2)	808 (11.9)	509 (9.7)
Conduct problems (*n*, %)	Male	3127 (66.6)	1027 (25.8)	725 (19.3)	596 (17.6)	289 (11.5)
	Female	2717 (61.3)	843 (22.2)	598 (16.1)	493 (14.4)	370 (13.6)
	Total	5844 (64.0)	1870 (24.1)	1323 (17.7)	1089 (16.0)	659 (12.6)
Emotional symptoms (*n*, %)	Male	1185 (25.0)	475 (11.9)	428 (11.4)	377 (11.2)	211 (8.4)
	Female	1178 (26.4)	525 (13.9)	575 (15.5)	472 (13.8)	510 (18.8)
	Total	2363 (25.7)	1000 (13.5)	1003 (13.6)	849 (12.5)	721 (13.8)
Peer relationship problems (n, %)	Male	---	647 (16.3)	665 (17.7)	598 (17.6)	432 (17.2)
	Female	---	463 (12.2)	520 (14.0)	472 (13.8)	385 (14.1)
	Total	---	1110 (14.3)	1185 (15.9)	1070 (15.7)	817 (15.6)
Prosocial behaviour (*n*, %)	Male	---	522 (13.1)	382 (10.2)	329 (9.7)	346 (13.8)
	Female	---	229 (6.0)	177 (4.8)	172 (5.0)	269 (9.9)
	Total	---	751 (9.7)	559 (7.5)	501 (7.3)	615 (11.8)

‘---’ Not measured/no data available. Total difficulties score range: 0–40; other domains score range: 0–10. Higher scores represent higher problems except for pro-social behaviour, where lower scores represent greater difficulties. Children were categorised as ‘borderline/abnormal’ if they scored in the ‘abnormal’ or ‘borderline ‘ranges, in accordance with cut-offs suggested by Goodman for total behavioural difficulties and for each SDQ subscale. Binomial cut-off: total difficulties (≥14), hyperactivity (≥6), conduct problems (≥3), emotional symptoms (≥4), peer relationship problems (≥3), pro-social behaviour (≤5).

**Table 3 behavsci-14-00049-t003:** Association between pre-pregnancy BMI and behavioural problems in children and adolescents over time (GEE model).

Pre-Pregnancy BMI (Kg/m^2^) and Gestational Weight Gain	Crude OR (95% CI)					
	Total Behavioural Difficulties	Hyperactivity/Inattention Problems	Conduct Problems	Emotional Symptoms	Peer Relationship Problems	Pro-Social Behaviours
Pre-pregnancy BMI						
<18.5	1.38 (1.16–1.64)	1.27 (1.01–1.61)	1.18 (1.02–1.35)	1.53 (1.27–1.83)	1.24 (0.98–1.58)	1.30 (0.98–1.71)
18.5–24.99	1	1	1	1	1	1
25–29.99	1.12 (1.01–1.24)	1.12 (0.98–1.29)	1.13 (1.05–1.23)	1.03 (0.92–1.15)	1.20 (1.04–1.37)	0.93 (0.77–1.10)
≥30	1.11 (0.94–1.32)	1.06 (0.84–1.33)	1.12 (0.99–1.28)	1.17 (0.97–1.39)	1.48 (1.20–1.82)	0.62 (0.44–0.86)
Gestational weight gain						
Inadequate	1.05 (0.96–1.15)	1.11 (0.98–1.25)	1.01 (0.95–1.08)	1.00 (0.91–1.11)	1.00 (0.89–1.13)	0.92 (0.80–1.07)
Adequate	1	1	1	1	1	1
Excessive	1.03 (0.94–1.13)	1.02 (0.90–1.17)	0.95 (0.88–1.02)	1.10 (0.99–1.22)	1.12 (0.99–1.27)	0.82 (0.70–0.96)
	Adjusted OR (95% CI) #					
Pre-pregnancy BMI						
<18.5	1.22 (1.02–1.45)	1.15 (0.91–1.47)	1.09 (0.95–1.26)	1.37 (1.14–1.64)	1.14 (0.90–1.46)	1.33 (1.01–1.75)
18.5–24.99	1	1	1	1	1	1
25–29.99	1.08 (0.98–1.20)	1.08 (0.94–1.25)	1.11 (1.03–1.21)	1.01 (0.90–1.14)	1.18 (1.03–1.37)	0.97 (0.81–1.15)
≥30	1.03 (0.87–1.21)	0.98 (0.77–1.23)	1.07 (0.94–1.22)	1.10 (0.92–1.32)	1.43 (1.16–1.77)	0.62 (0.44–0.86)
Gestational weight gain						
Inadequate	1.04 (0.96–1.14)	1.09 (0.97–1.24)	0.99 (0.92–1.06)	1.00 (0.91–1.11)	1.01 (0.89–1.14)	0.91 (0.79–1.05)
Adequate	1	1	1	1	1	1
Excessive	0.96 (0.87–1.05)	0.96 (0.84–1.09)	0.92 (0.85–0.99)	1.03 (0.93–1.14)	1.06 (0.94–1.20)	0.82 (0.70–0.96)

# Adjusted for maternal age, education, marital status, parity, maternal smoking, alcohol use, anxiety and depressive symptoms during pregnancy, and offspring sex.

## Data Availability

The data used in this study are available from the corresponding author on reasonable request and ASLPAC executive committee approval.

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
