# Peer review of "Maternal Pregnancy and Pre-Pregnancy Weight and Behavioural Outcomes in Children"

_behavsci, 2024, doi:10.3390/bs14010049_

Round 1

Reviewer 1 Report

Comments and Suggestions for Authors

This manuscript examines the role of pre-pregnancy BMI and gestational weight gain on offspring behavioural problems at multiple time points in childhood. It is a significant contribution, due to the discordant data on the effects of maternal BMI on offspring behavioural problems, especially due to the large sample size. There are also not many studies that examine the role of maternal underweight, gestational weight gain, and outcomes besides hyperactivity-inattention problems in children. However, what makes this study a very significant contribution is the long-term follow-up and multiple evaluations at various neurodevelopmental stages. But, I do have some significant concerns about the study population and adjustment factors. 

Major Concerns
My biggest concern is about the exclusion of any children not born at term and not born with a “normal” weight. First, and least importantly, what you consider as “at term” and “normal” weight are not defined. Second, your reason for this exclusion is not stated. Third, and most importantly, I see no reason for this exclusion. Maternal BMI is associated with both gestational age and weight, which are associated with offspring neurodevelopment. You are excluding a whole portion of your study sample on certain conditions of mediators and biasing your results. It is not common epidemiological process to condition your analyses on mediators. At a very minimum, if you do wish to do this, you should provide a compelling argument supporting this exclusion and present analyses both with and without this exclusion.

My next concern is in regards to your adjustment for urinary tract infections. Why are you adjusting analyses for UTIs? It seems to me that UTIs are more prevalent with obesity, but then this would mean that UTIs are also a mediator and you should not be adjusting for it in your analyses.

My other points are more minor, I've arranged them in order of appearance in the manuscript.

Introduction
I don’t think you need to start the paragraph defining BMI, it starts the reader off with a tedious tone and you define it later in the methods.

In the second paragraph, « existing studies show that only a few pregnant women achieved weight gain »… This sounds very strange, it sounds as if you’re citing very tiny studies, it would be better to cite a percentage or a range of women. For example, “existing studies show that only 20-30% of women achieve weight gain in the recommended range…”. I would also put the sentence in the present tense, as it would be a trend you’re describing. 

Methods
Mothers reported their pre-pregnancy weight and height in early pregnancy. When? State this specifically using a range or an average time. Same comment for the first antenatal clinical visit.

Add units for the WHO categories.  

When describing the SDQ, the category should be “hyperactivity-inattention”, not just hyperactivity.

In the flowchart and in the study participants section, you state that you only included those without any missing values for the exposure, outcome or covariates. But in the statistical analysis section, you describe the multiple imputation technique you used. After reading the whole manuscript it becomes clearer that the imputed analyses were sensitivity analyses, but this is not clear in the methods. Please clarify. It is also standard practice to use the imputed analyses as the main results, not the complete-cases, which would bias your results due to missing data. The complete-case analysis is usually presented in the supplementary material. Also, according to MI reporting guidelines, a supplementary table describing your missing data should be provided. 

Results
Table 1. Some formatting errors – no spaces between the mean and SD or N and %.

Instead of titling your category n (%) and then putting mean, SD on every line for continuous variables, why not just title the category N (%) or Mean [SD]?

For the levels of education, I think that most people outside of the UK will not know what your O and A levels correspond to. Could you change the categories or names so that international readers will have a better idea?

How were maternal anxiety and depressive symptoms determined?

Is the “infection during pregnancy” variable only referring to urinary tract infections? In the Methods you only describe urinary tract infections.

You state “Broadly consistent results were obtained when we reran the analyses using the continuous EPDS scores”. What are you referring to? Did you mean to say the SDQ scores?

You reference Table 4 in the results, but I think you mean table S4.

I’m confused about the text in the section for “maternal vs paternal BMI”. You cite an OR=1.36 in Table S5, but your table S5 is titled pre-pregnancy obesity and you present beta coefficients. I am also confused about what your Table S6 is for, are these supposed to be the maternal BMI adjusted for paternal BMI? It is not stated, but in the text, you say that this table is supposed to present the maternal BMI ORs adjusted for paternal BMI.

Are you adjusting the gestational weight gain models for paternal BMI as well? This is also not clear.

What is the COR in Table S7? Crude OR? And AOR is the adjusted? Needs to be defined. The units for BMI must also be added.

The text in the whole section for maternal vs paternal BMI does not correspond with the tables. The whole section is highly confusing.

There are too many supplementary tables and they are redundant. I think the last 2 tables can be removed completely and the findings cited in the text. However, I think the authors should remove more than those two tables.

Table 2.
Unless I’m mistaken, you don’t mention anywhere in the Methods that you tested for an interaction with offspring sex, but you present the stratified results. Also, why did you present the sex-stratified results if you concluded that the interaction is not significant (though many might argue that for interactions, a threshold p<0.10 is significant). Normally, we don’t present stratified results if the interaction is not significant nor if we do not have an a priori reason for doing so (which was not mentioned in your methods).

Perhaps, to make this table easier to read, you could delete all the columns with your p-value and use a symbol instead to mark the significant differences between males and females. 

Table S1.
Should be hyperactivity-inattention, not just hyperactivity.

Discussion
You state that you are not aware of any other studies that examine the association between pre-pregnancy BMI and behaviour at multiple time points, however, here is one: Dow, C., Galera, C., Charles, MA. et al. Maternal pre-pregnancy BMI and offspring hyperactivity–inattention trajectories from 3 to 8 years in the EDEN birth cohort study. Eur Child Adolesc Psychiatry 32, 2057–2065 (2023). https://doi.org/10.1007/s00787-022-02047-x

You also state only being aware of Jo’s study with regards to underweight, here are others, in addition to the above reference already provided: Maternal pre-pregnancy obesity and offspring hyperactivity-inattention symptoms at 5 years in preterm and term children: a multi-cohort analysis
doi: 10.1038/s41598-022-22750-8

High maternal pre-pregnancy BMI is associated with increased offspring peer-relationship problems at 5 years
Doi : 10.3389/frcha.2022.971743

 You state that you are only aware of 2 other studies examining the role of GWG on offspring behavioural development, here is another: Prepregnancy body mass index and gestational weight gain affect the offspring neurobehavioral development at one year of age
 PMID: 33832396
 DOI: 10.1080/14767058.2021.1907336

You say that you adjusted for a wide-range of confounders, including pre-eclampsia, but you don’t state it as an adjustment factors in any of the tables and adjustment for pre-eclampsia would be erroneous because pre-eclampsia could be considered a mediator.

What about adjusting for maternal diet and physical activity during pregnancy? These are both very important confounders in the relationship between maternal BMI and offspring neurodevelopment. I am pretty sure that ALSPAC has at least dietary information available.

With regards to attrition, why not use a method to try to reduce the bias? Such as inverse probability weighting?

Overall
Formatting mistakes all throughout the manuscript, especially with not leaving spaces between the reference and the last word before the reference. Author names are misspelled in the discussion at least twice. Numbers in the figures and tables should be formatted correctly (ie. 14,541 and not 14541). Arrows are not correctly positioned in Figure 1 and there is an extra, empty bullet point in Figure 1. 

Comments on the Quality of English Language

Minor grammatical errors to be corrected throughout the manuscript. Misspelling of author names in the discussion.

Reviewer 2 Report

Comments and Suggestions for Authors

This study, utilising data from the Avon Longitudinal Study of Parents and Children (ALSPAC), investigates the associations between pre-pregnancy weight, gestational weight gain, and offspring behavioural outcomes at five developmental stages from ages 3 to 16. It includes a comprehensive analysis covering various developmental stages, a clear presentation of associations between maternal factors (BMI and gestational weight gain) and child behavioural outcomes, and an exploration of paternal BMI that adds depth to understanding familial influences.

Suggestions to authors:

·        Consider providing additional contextual information, such as the socioeconomic status or racial/ethnic diversity of the study population, which could influence the generalisability of the results.

·        When associations observed in univariate models change in the adjusted models, provide potential explanations or hypotheses for the observed changes, considering the role of confounding variables.

·        Explicitly discuss limitations, including potential biases in self-reported weight and height data.

·        Discuss the practical significance of the observed effect sizes and the implications of the findings for public health or clinical practice. How could the observed associations inform interventions or policies?

·        While the paper acknowledges and adjusts for several confounding factors during pregnancy, it is crucial to consider other potential confounders that might influence child behaviour during growth and development, including parenting style, socioeconomic status, and environmental stressors. Additionally, consider discussing genetic, familial, and mental health factors, emphasising the importance of extending confounder consideration beyond the prenatal period.

·        In the conclusion, include the implications of the main findings. Consider discussing the significance of the weak evidence of associations between pre-pregnancy overweight, obesity, and gestational weight gain and child behavioural outcomes.

Reviewer 3 Report

Comments and Suggestions for Authors

This is a very interesting paper investigating the impact of maternal pregnancy and pre-pregnancy weight on clinical outcomes in children born from these women. The paper is well written and of interest for the readers; however, several minor changes are recommended before considering it for publication.

ABSTRACT

1- How were the patients recruited in the study? This should be explained in detail in abstract.

2-How many patients were recruited?

INTRODUCTION

1- The authors have described that several metabolic variables, including weight have been associated with clinical outcomes in children, for instance behavioral disturbances. I recommend to add the explanation of other variables that should be controlled when analyzing results. It is well established fact that there are other genetic and environmental factors capable of influencing the brain of children of exposed women. These confounding variables should be included in the introduction section.

METHODS

1- Please, provide a reference from the ALSPAC cohort study, if this is not the first report of the study.

RESULTS

Main characteristics of the sample and analysis according to weight, etc, should be described separately. I recommend to divide the results section into several subsections.

DISCUSSION

The conclusions should be stated in a separate section called "Conclusions".
